# Statistical mechanics for metabolic networks during steady state growth

Daniele De Martino[1], Anna MC Andersson[1], Tobias Bergmiller [1], Călin C. Guet[1] & Gašper Tkačik [1]

Which properties of metabolic networks can be derived solely from stoichiometry? Predictive results have been obtained by flux balance analysis (FBA), by postulating that cells set metabolic fluxes to maximize growth rate. Here we consider a generalization of FBA to single-cell level using maximum entropy modeling, which we extend and test experimentally. Specifically, we define for *Escherichia coli* metabolism a flux distribution that yields the experimental growth rate: the model, containing FBA as a limit, provides a better match to measured fluxes and it makes a wide range of predictions: on flux variability, regulation, and correlations; on the relative importance of stoichiometry vs. optimization; on scaling relations for growth rate distributions. We validate the latter here with single-cell data at different sub-inhibitory antibiotic concentrations. The model quantifies growth optimization as emerging from the interplay of competitive dynamics in the population and regulation of metabolism at the level of single cells.

---

[1] Institute of Science and Technology Austria, Am Campus 1, A-3400 Klosterneuburg, Austria. Correspondence and requests for materials should be addressed to D.D.M. (email: ddemarti@ist.ac.at)

After the significant developments in molecular biology and biochemistry in the last century, many aspects of cellular physiology could be understood as a result of interactions between identified molecular components. Perhaps the best-characterized example is intermediate metabolism, the set of reactions that enable cell growth by converting organic compounds and transducing free energy. Today it is possible to some extent to infer the topology of metabolic networks from data at genomic scale, but the dynamics and parameter dependence of such networks remain difficult to analyze. Alternatively, one can assume that known reactions only provide physico-chemical constraints within which some adaptive dynamics has maximized the growth rate, e.g., by adjusting enzyme levels and controlling reaction rates[1]. An influential implementation of this idea for batch cultures under steady-state conditions has been the flux balance analysis (FBA)[2], which has been tested experimentally[3,4], also in mutant strains, strains used for industrial production[5–7], as well as phenotypes implicated in disease (e.g., Warburg effect[8]). Using maximum entropy ideas from statistical physics, we extend the application of FBA from batch to single-cell level and show that our extension makes a wide range of predictions, some of which we test experimentally.

Recent measurements at the single-cell level demonstrated the existence of substantial cell-to-cell growth rate fluctuations even in well-controlled steady-state conditions[9]. These fluctuations exhibit universal scaling properties[10–13], relate to cell size control mechanisms[14,15], act as a global collective mode for heterogeneity in gene expression[16–18], and are ultimately believed to affect fitness[19]. To link these observations to metabolism, however, we need to set up a mathematical description not only of the optimal metabolic fluxes and maximal growth rate in batch culture (as in FBA, which permits no heterogeneity across cells), but for the complete joint distribution over metabolic fluxes. Metabolic phenotypes of individual cells growing in steady-state conditions can then be understood as samples from this joint distribution, which would automatically contain information about flux correlations, and, in particular, could directly predict cell-to-cell growth rate fluctuations.

The simplest construction of a joint distribution over metabolic fluxes can be derived in the maximum entropy framework[20]. The key intuition is to look for the most unbiased (or random) distribution over fluxes through individual metabolic reactions that is consistent with the given stoichiometric constraints, while matching the experimentally measured average growth rate. The maximum entropy model that we specify below will turn out to be a one-parameter family of distributions, where the single parameter can be fit to match experimental data; all subsequent predictions follow directly, without further fitting. A similar approach has recently been used in diverse biological settings, ranging from neural networks[21,22], genetic regulatory networks[23], antibody diversity[24], and collective motion of starling flocks[25].

In addition to accounting for cell-to-cell variability, the maximum entropy construction provides a principled interpolation between two extremal regimes of metabolic network function. In the uniform (no-optimization) limit, no control is exerted over metabolic fluxes: they are selected at random as long as they are permitted by stoichiometry, resulting in broad yet non-trivial flux distributions that support a small, non-zero growth rate. In the FBA limit, fluxes are controlled precisely to maximize the growth rate, with zero fluctuations. The existence of these two limits defines a fundamental, and still unanswered, question about metabolic networks: Is there empirical evidence that real metabolic networks are located in an intermediate regime between the two limits where fluctuations are non-negligible[26], and if so, what are the properties of this intermediate regime (see Fig. 1)? Here,

we address this question using metabolic flux and single-cell physiology data for *Escherichia coli*.

In the Methods section, we provide a review of the maximum entropy formalism for metabolic networks as it has been established in previous work[26–31] and we stress its implementation in our work. In the Results section, we use this formalism to set up and test quantitative predictions for *E. coli* as well as to discuss possible theoretical extensions. We provide here compelling experimental evidence that the observed growth rate fluctuations reflect metabolic flux variability and sub-optimality of growth, both of which are captured quantitatively by the maximum entropy model of the metabolic network.

## Results

**Experimental test of flux predictions for *E. coli*.** We constructed a maximum entropy model for the catabolic core of the *E. coli* metabolism (see Methods section). The model has a single parameter $\beta$ that constrains the average growth rate in the flux space, interpolating from an uniform sampling ($\beta = 0$) to the optimal FBA solution ($\beta \to \infty$). In particular, we consider the specific value of this parameter $\beta^*$ inferred by constraining the average growth rate in the model to match the population experimental growth rate ($\bar{\lambda} = 0.2\,\mathrm{h}^{-1}$ for a set of 12 experiments shown in Fig. 2a, and $\bar{\lambda} = 0.1\,\mathrm{h}^{-1}$ for a set of seven experiments, not shown).

To evaluate the quality of model predictions, we compared $N_f = 20$ measured metabolic fluxes in *E. coli* from previously published data to our predictions, as shown in Fig. 2a. We defined the mean-squared error (MSE) as

$$\mathrm{MSE} = N_f^{-1} \sum_{i=1}^{N_f} (\langle v_i \rangle - V_i)^2, \qquad (1)$$

where $V_i$ is the measured flux (relative to glucose uptake) and $\langle v_i \rangle$ is the mean of the corresponding flux computed in the maximum entropy model of Eq. (7). Figure 2b examines the behavior of MSE as a function of the parameter $\beta$. First, we note that the best flux predictions occur at or close to the value $\beta^* \lambda_{\max} \simeq 120$, identified by the maximum entropy fit, at both average growth

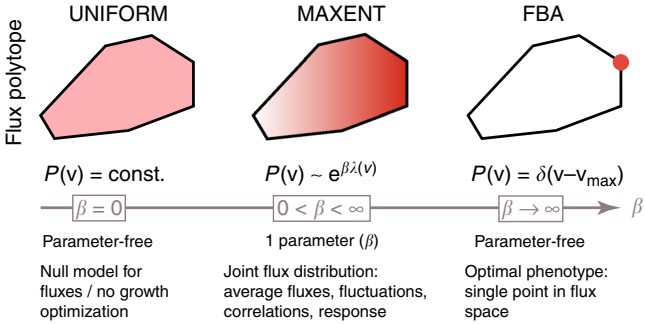

**Fig. 1** Stoichiometric and thermodynamic constraints define a high-dimensional convex polytope of permissible fluxes, here shown in cross-section schematically as a black polygon. In a uniform model, the flux distribution, $P(\mathbf{v})$, is uniform over this polytope (UNIFORM). In contrast, flux balance analysis (FBA) finds a single permissible and optimal combination of fluxes, $\mathbf{v}_{\max}$ (red dot), such that the growth rate is maximal, $\lambda_{\max}$. FBA and the uniform model are two limits (of $\beta \to \infty$ and $\beta = 0$, respectively) of a one-parameter family of distributions (MAXENT), where increasing the parameter $\beta$ biases the flux distribution (red gradient) away from uniform toward achieving higher average growth rates, $\bar{\lambda}(\beta)$. The distribution over fluxes has a Boltzmann form from statistical physics and corresponds to a case where fluxes are as random as possible while achieving a specified growth rate

rates. This is a non-trivial prediction, because the value of $\beta^*$ was not fitted to minimize MSE, but rather, as demanded by the maxent formalism, to match the population growth rate. Second (and unsurprisingly), we find that flux predictions are better at $\beta^*$ than with uniform sampling, at $\beta = 0$. Perhaps the most surprising is our third finding: flux predictions at intermediate value of beta ($\beta^*\lambda_{max} \simeq 120$) significantly outperform the limit of $\beta \to \infty$, i.e., the FBA solution.

In addition to a better quantitative match overall, the maximum entropy model correctly predicted non-zero flux through the glyoxylate shunt, i.e., for the isocitrate lyase (ICL) as well as ME1 reactions, which FBA misses qualitatively by setting them to zero. As a consequence, this also leads to a better match of our model with data for reactions isocitrate dehydrogenase (ICDH) and alpha ketoglutarate dehydrogenase (AKGDH) that channel pyruvate through the Krebs cycle.

Lastly, we point out that Eq. (7) can also be viewed as a phenomenological equation for average fluxes with a single fitting parameter $\beta$ that is set not to match the measured population growth rate, as in the maxent formalism, but simply to minimize some error measure (say MSE) with respect to experimentally measured fluxes. In Supplementary Note 4, we show that this also leads to predictions that outperform FBA for measured fluxes both in wild-type and mutant strains.

It is instructive to examine the evolution of the joint distribution over fluxes, $P_\beta(\mathbf{v})$, as a function of the optimization parameter, $\beta$. Figure 3a shows how the growth rate approaches the maximal rate achievable, $\lambda_{max}$, with the inferred values of $\beta^*$ from Fig. 2a, c suggesting an optimization level in the range of ~80% of the maximum. These levels are reached by adjusting flux values away from what they would have been under uniform sampling from the polytope of the allowed metabolic phenotypes, $\mathcal{P}$. Figure 3b traces the relative changes in all fluxes as a function of $\beta$. Interestingly, in the FBA limit, almost half of the fluxes (38 out of 86 fluxes, the upper half of the plot) are forced to zero, whereas at the inferred value ($\beta^*\lambda_{max} \approx 120$), these fluxes only decrease by about 1/3 relative to their average value in the uniform sampling limit. Furthermore, the glyoxylate shunt remains active, in agreement with experimental observations.

Surprisingly, only for a few reactions the fluxes are predicted to increase with growth rate optimization relative to the uniform sampling (lowest ~5 fluxes in Fig. 3b). These are mainly nitrogen and phosphate transport reactions, and to a lesser extent, malate dehydrogenase (MDH) and phosphoglucose isomerase (PGI) reactions. The latter two reactions are classified as reversible, whereas regulation in metabolic networks is thought to take place at irreversible reactions[32], so the predicted increase may be a consequence of increased substrate levels.

We separately illustrate three flux behaviors in Fig. 3c, for ICL, ICDH, and glutamate dehydrogenase (GLUDy). ICL and ICDH track the relative channeling of carbon sources in the Krebs cycle vs. glyoxylate shunt; ICL flux is switched off in the $\beta \to \infty$ limit, whereas ICDH flux remains nearly constant with $\beta$. In contrast, GLUDy reaction is reversible, switching sign at intermediate values of $\beta$, while at high $\beta$ the reaction ultimately gets frozen in the backward direction, implying high levels of ammonia in the cell, given the low affinity of this enzyme for ammonia[33].

We also evaluated the predicted variability in metabolic fluxes from the maximum entropy model, Eq. (7), at $\beta^*\lambda_{max} \sim 10^2$, and found a clear division between reactions with high and low coefficients of variation (CV). Among fluxes with lower variability were all glycolytic reactions (CV < 0.3) with the exception of PGI, as well as all transport reactions related to biomass formation (i.e., for glucose, oxygen, ammonia, carbon dioxide, phosphate ions; CV < 0.11), the first part of the Krebs cycle, and the irreversible reactions of oxidative phosphorylation.

We next wondered how flux variances scale with the optimization level. In the uniform sampling limit ($\beta = 0$), the variances should be large, characteristic of the shape and extent of the permitted polytope, $\mathcal{P}$. While in the FBA limit ($\beta \to \infty$) the flux variability should vanish, we expect a well-defined scaling regime at high $\beta$ where the variances shrink toward the FBA solution in a manner that is independent of the global polytope properties. This regime is indeed reached for all fluxes at $\bar{\lambda}/\lambda_{max} \gtrsim 0.90$ and for some fluxes much earlier, as shown in Fig. 3d: flux variability subsequently decreases with $\beta$ as $\sigma_i(\beta)/\sigma_i(\beta = 0) \propto (1 - \bar{\lambda}(\beta)/\lambda_{max})$.

What kind of correlation structure between fluxes does the maximum entropy model predict? While the growth rate $\lambda$ is

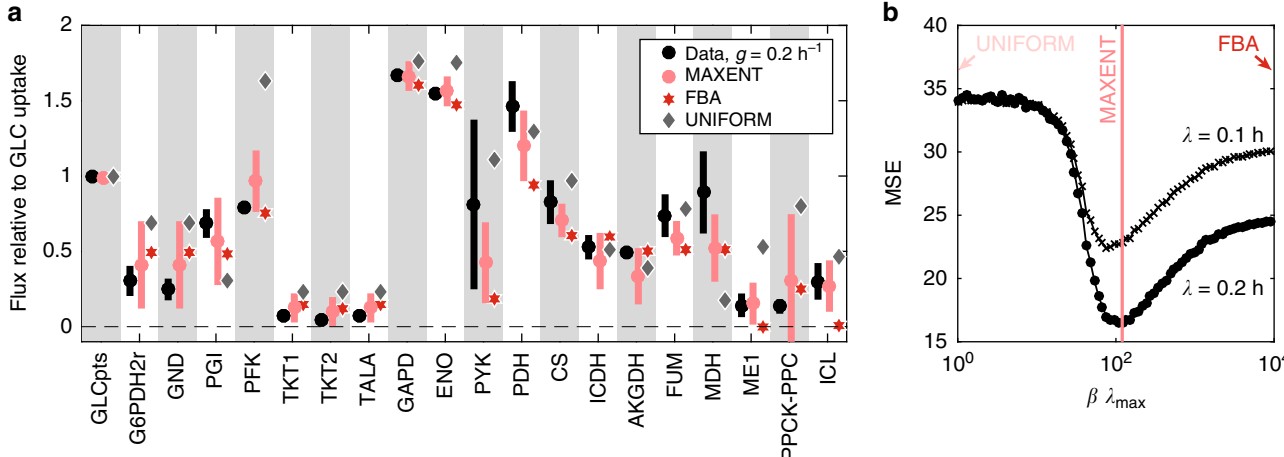

**Fig. 2 a** Comparison of measured fluxes (black, mean, error bars defined as SD over 12 experiments, technical replicates; normalized to glucose uptake) with predictions of FBA (red stars) and of the maximum entropy model (pink, error bars defined as SD with simulation sample size $10^5$). Also shown are mean fluxes predicted by uniform sampling, i.e., using $\beta = 0$ in Eq. (7) (gray stars; mean, for clarity, large SDs are not displayed). Data for **a** are a collection of 12 experiments at average growth rate $\bar{\lambda} = 0.2\ h^{-1}$. Wild-type *E. coli* was grown in glucose-limited medium in aerobic conditions. **b** Quality of flux predictions, quantified as mean-squared error or MSE in Eq. (1) between flux measurements and the maximum entropy predictions, as a function of dimensionless $\beta\lambda_{max}$ parameter. Two curves correspond to models inferred at two different average growth rates, $\bar{\lambda} = 0.2\ h^{-1}$ (data from **a**) and $\bar{\lambda} = 0.1\ h^{-1}$ (Supplementary Data 1). Maximum entropy models for both growth rates coincide at an intermediate value of $\beta^*\lambda_{max} \approx 120$ (pink line) and provide a better fit than the uniform ($\beta = 0$) or FBA ($\beta \to \infty$) limits

linear in constituent fluxes in Eq. (7), suggesting that the joint distribution could factorize, correlations between fluxes develop because of the stoichiometric constraints that define the polytope $\mathcal{P}$. A subset of fluxes that we focus on in Fig. 3 exhibits a clear structure of strong (anti-)correlation both under uniform sampling (Fig. 3e) and in the FBA limit (Fig. 3f). The FBA pattern of correlations, in particular, can easily be partitioned into four groups using a clustering algorithm[34] so that the groups are strongly enriched for reactions characteristic of glycolysis, glyoxylate shunt, pentose phosphate pathway, and citric acid cycle, respectively. Fluxes in the glycolysis cluster tend to correlate strongly with fluxes in the citric acid cycle cluster, but anti-correlate with glyoxylate shunt and pentose phosphate pathway cluster. Comparison of the FBA correlations (F) with the uniform sampling (E) reveals that stoichiometric constraints alone shape much of the correlation structure, with the exception of anti-correlation between glycolysis and glyoxylate shunt clusters, which is a distinct consequence of the growth rate optimization. More generally, it is intriguing to apply maximum entropy to recover the correlation structure of metabolic fluxes in the FBA limit and use that to identify, automatically via clustering, separate metabolic pathways.

**Lower limit of regulatory information needed for fast growth.** As the growth rate optimization parameter $\beta$ is increased, flux variances shrink (Fig. 3d), correlations strengthen (Fig. 3f), and the distribution over fluxes within the polytope $\mathcal{P}$ localizes closer to the FBA solution, $\mathbf{v}_{max}$. Could this localization emerge due to

competitive growth dynamics in batch culture[28]? To test this hypothesis, we checked the relationship predicted by Eq. (13), which can be solved analytically (see Supplementary Methods). For the typical values of the carrying capacity vs. inoculation size ratio ($N_C/N_0 \simeq 10^6$), the corresponding prediction from Eq. (13) is $\beta^* \lambda_{max} \sim 50$, considerably underestimating $\beta^* \lambda_{max} \simeq 120$, recovered by the maximum entropy model as reproducing the population growth rate and showing good match with measured fluxes. In other words, metabolic fluxes are more localized and growth is closer to optimal than would be expected in a scenario where metabolic reactions at the individual cell level are not actively regulated.

Motivated by these findings, we explored an alternative scenario where the localization of the distribution over fluxes around the optimal growth rate is achieved by active regulation of metabolic reactions. First, we quantified the degree of localization by the decrease in the entropy of the distribution over fluxes, Eq. (10). This is plotted for our *E. coli* network in Fig. 3g, where we show the average growth rate, $\bar{\lambda}$, as a function of information, $I$ (expressed in bits), parametrically in $\beta$. The resulting curve divides the $(I, \bar{\lambda})$ plane into two halves: while it is possible to achieve metabolic phenotypes below the $I(\bar{\lambda})$ curve, the dashed region above the curve is forbidden. This is because *no* distribution exists that achieves high growth rates $\bar{\lambda}$ without also deviating from the uniform distribution by at least the required number of bits.

Figure 3g suggests that at least ~40 bits of information are required to control the fluxes and reach growth rates amounting

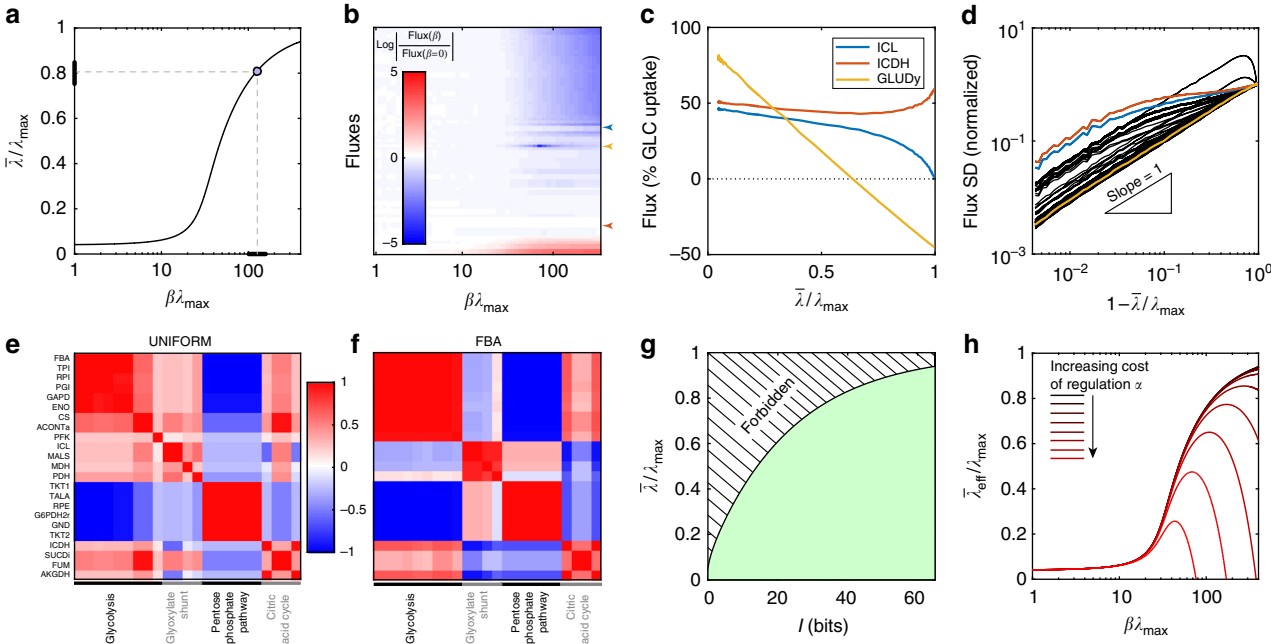

**Fig. 3 a** Average growth rate (relative to maximal achievable rate) as a function of $\beta\lambda_{max}$. Inferred value for data in Fig. 2a is shown as a blue point (thick black line = STD for $\bar{\lambda}/\lambda_{max}$ and the corresponding range for $\beta\lambda_{max}$). **b** Log fold change of average fluxes as a function of $\beta\lambda_{max}$, relative to the uniform distribution; fluxes are sorted by their change in the FBA ($\beta \to \infty$) limit. **c** Dependence of three selected fluxes (legend, highlighted with correspondingly colored arrows in **b**) on the average growth rate, $\bar{\lambda}$: ICL is turned off in the FBA limit (but not in the maximum entropy solution), ICDH remains nearly unchanged with $\bar{\lambda}$, and GLUDy is the only flux that switches sign. **d** Flux variabilities (each black line = SD of one flux according to maximum entropy distribution, three selected fluxes from **c** highlighted in color) scale linearly with distance to maximal growth rate and vanish in the FBA limit. **e**, **f** Correlation coefficient matrix (Pearson correlation, color scale) between 23 selected fluxes in the uniform (**e**) and FBA (**f**) limits, computed within maximum entropy framework. Fluxes have been grouped into four clusters according to the correlation in the FBA limit and reordered accordingly in both plots; clusters are strongly enriched for fluxes belonging to pathways denoted at bottom. Note the flip in correlation sign between the glycolysis and glyoxylate shunt pathways between the two limits. **g** Achieving a particular growth rate (y-axis) requires reducing the entropy of the joint distribution of fluxes at least by $I$ bits below the entropy of the uniform distribution (green region). Points in the hashed (forbidden) region are not achievable. **h** A simple model in which tight regulation of fluxes (higher $I$) enables higher growth rates, as in **g**, but also entails metabolic cost (see text). For a given cost $\alpha$, the effective growth rate $\bar{\lambda}_{eff}$ is maximized at an intermediate value of $\beta\lambda_{max}$

to ~80% of the maximal rate, $\lambda_{max}$, as reported in data for *E. coli*; higher growth rates call for increasing amounts of information, which formally diverges in the FBA limit as $\beta \to \infty$. Interestingly, the number of out-of-equilibrium reactions in the model, 39, is in good agreement with the inferred amount of minimal information, given a simplistic estimate of 1 bit per reaction (sufficient to distinguish, e.g., high from low expression of the metabolic enzyme). This is consistent with the hypothesis that regulatory control is exerted for enzymes that catalyze irreversible reactions[32].

Cells control metabolic fluxes through regulatory networks, either indirectly, by regulating the expression of metabolic enzymes, or directly, by modulating the enzymatic activity through various feedback loops; either way, metabolic resources are required to exert this control. This leads to a trade-off: flux control is necessary to support a high growth rate, but itself carries a growth rate penalty. We created a simple toy model to capture this intuition (see Supplementary Note 2). Here, $K$ regulatory pathways control the fluxes and each pathway is modeled as a Gaussian information channel, so that together, these channels provide $I(\beta)$ bits of necessary information as shown in Fig. 3g. The signal-to-noise of each regulatory channel is determined by the number of regulatory molecules: higher molecular counts enable precise control and thus higher information, but impose higher cost. In this model, the cost-free growth rate at given $\beta$ is reduced by the cost to support $K$ channels which control the fluxes, so that the resulting effective growth rate is:

$$\bar{\lambda}_{\text{eff}}(\beta) = \bar{\lambda}(\beta) - \alpha K \left[ 2^{2I(\beta)/K} - 1 \right], \qquad (2)$$

where $\alpha$ determines the metabolic cost of regulatory molecules, and we estimated the number of regulatory pathways, $K$, to be approximately the number of degrees of freedom of the flux polytope, $K \approx D$. The cost of regulation clearly limits the achievable growth rate, as shown in Fig. 3h, where the $\bar{\lambda}_{\text{eff}}(\beta)$ curves now develop a maximum rather than increasing monotonically as in the cost-free case of Fig. 3a. While our toy model is very simplistic, it does capture properly the scaling of information with the growth rate, as well as the exponential metabolic cost of achieving high information transmission in molecular networks, reported previously[35,36]. Thus, among many possible constraints acting on a cell, the cost of regulating metabolism itself[1] can impose non-negligible limits to growth.

**Experimental test of growth rate fluctuation scaling**. Can we test the novel predictions of our theory that extend beyond the domain of validity of the FBA? While it is currently experimentally unfeasible to measure metabolic fluxes and their variability at the single cell level, one can tractably measure division times and growth rates for single *E. coli* cells growing in stable conditions for long periods of time. In our model, such growth measurements directly connect to the biomass producing reaction with its associated growth flux $\lambda(\mathbf{v})$. Figure 3d suggests that flux variability should scale $\propto (\lambda_{max} - \bar{\lambda})$, and since the growth flux is a linear combination of metabolic fluxes, its variability, too, should follow the same scaling. To verify this explicitly, we computed the fluctuations in growth rate, $\sigma/\lambda_{max}$, as a function of the optimization parameter $\beta$, in Fig. 4a.

In the range of $\beta\lambda_{max} \gtrsim 40$, characteristic of wild-type *E. coli* experiments, the predicted growth fluctuations indeed obey

$$\frac{\sigma}{\lambda_{max}} \propto (\beta\lambda_{max})^{-1} \propto (\lambda_{max} - \bar{\lambda}); \qquad (3)$$

we refer to this range as the scaling regime. Beyond variance, the

complete distribution of growth rates, $Q(\lambda)$, can be sampled by marginalizing the maximum entropy model, Eq. (7).

Measurements of single-cell growth rates allow us to estimate growth rate distributions and compare them to the predicted $Q(\lambda)$, as well as to empirically extracted $\bar{\lambda}$, $\lambda_{max}$, and the fluctuations $\sigma$, to verify the predicted relation of Eq. (3). We used previously published data[37] where *E. coli* cells were stably grown in a mother machine microfluidic device while multiple sub-inhibitory steps of concentration of the antibiotic tetracycline were delivered as shown in Fig. 4b. Low concentrations of antibiotic allowed us to probe different average growth rates in the same setup, and to construct empirical distributions of growth rates for every antibiotic concentration by pooling data from technical replicates of the multi-step experiments (Supplementary Note 3). We find an excellent match between measured and predicted growth rate distributions in Fig. 4c for all five concentrations of the antibiotic used. Looking at many individual lineages in separate microfluidic channels, we can also extract $\lambda_{max}$, $\bar{\lambda}$, and $\sigma$ per lineage empirically and confirm the predicted scaling of growth rate fluctuations, as shown in Fig. 4d.

To conclude this section on fluctuations, we briefly mention that, under mild conditions, it is possible to study the dynamical response of the network in the linear regime under small perturbations. For example, ref.[26] introduced a simple biologically motivated dynamics of diffusion-replication inside the metabolic space, described by the one-parameter equation for the growth rate distribution $Q(\lambda)$, showing that the following fluctuation-dissipation scaling laws should hold for the typical response times as well as the growth rate fluctuation autocorrelation time $\tau$:[29]

$$\sigma \sim \left( \lambda_{max} - \bar{\lambda} \right) \sim \tau^{-1}. \qquad (4)$$

This relation, which further extends Eq. (3), is experimentally testable and predicts a divergent slowing down of the response time with growth rate maximization. As a consequence, growth rate fluctuations could take on a functional role in speeding up the response to environmental perturbations, e.g., to nutritional up-shifts or externally applied stresses. Even if the experimental test of such dynamic predictions is beyond the scope of this paper, our model connects to a wide range of currently ongoing metabolism- and growth-related investigations.

## Discussion

In this work, we considered maximum entropy distributions at fixed average growth rate in the space of metabolic phenotypes, a straightforward and statistically rigorous extension of the FBA, which is recovered in the asymptotic limit. Experimental estimates of enzymatic fluxes of the central carbon core metabolism in bulk cultures of *E. coli*, as well as empirical growth rate distributions of *E. coli* collected from single cell measurements, are consistent with intermediate level of growth optimization ($\beta\lambda_{max} \sim 10^2$ and $\bar{\lambda}/\lambda_{max} \sim 0.8$). We find that variability can be captured by a simple maximum entropy model, and that the zero-fluctuation FBA limit qualitatively misses important experimental facts, e.g., the observed non-zero fluxes through the glyoxylate shunt.

The improved ability of our model to match the flux measurements is a consequence of a single extra parameter, $\beta$, which can easily be determined from existing experimental data. Beyond a better fit, however, our model also makes a wide range of predictions, extending the domain of metabolic network analysis to the single-cell level. While it is difficult to measure the single-cell metabolic fluxes and their variability in isogenic populations in steady state, such measurements for the growth rate are increasingly available. This connection enables the new

predictions of our theory to be tested, and opens up the theory for verifiable extensions. Validating the predicted scaling of growth rate fluctuations in Fig. 4 is only the first step, with two broad lines of investigation within reach.

First, our approach is not limited to the core catabolism analyzed here or to bacterial metabolism, but can in principle be extended to other genome-scale networks. In practice, however, we often lack suitable large-scale experimental flux measurements. It is also likely that physico-chemical constraints alone are insufficient to yield quantitatively accurate predictions. Similar issues arise also in the core catabolism for high growth rates that exceed the threshold of the acetate switch, for which a trade-off between growth yield and rate emerges and additional constraints have to be added in FBA-based approaches[38]. Our method can be extended to accommodate such cases, or systems where strict growth maximization is likely not a suitable objective. Extra objectives or constraints in the maximum entropy would appear as additional terms in the exponent of Eq. (7), where their corresponding parameters would control various trade-offs between the objectives. This flexibility may be required to model metabolic dependencies, cell type heterogeneity, or interactions between cells.

Such extensions of maximum entropy modeling will benefit from the recent flourish of statistical-physics-inspired algorithms, ranging from belief propagation[39–41], relaxational learning[42], and gaussian analytical approximation[43], used to solve the sampling problem, which is computationally at the heart of our approach (where on the other hand FBA relies on simpler linear programming). While the employed Monte Carlo hit-and-run Markov chain is sufficient for the analyzed network, faster methods (in particular[43]) will pave the way to large-scale applications and inverse modeling settings.

Second, we considered two possible mechanisms for the emergence of maximum entropy distributions over metabolic fluxes. On the one hand, analysis of population dynamics of competitive growth under resource constraints reveals that maximum entropy distributions at fixed average growth rate are the steady states of logistic growth, in principle giving further testable predictions on the dependence of the growth optimization from inoculum size and medium carrying capacity. In essence, it is the exponential character of the growth laws that leads naturally to Boltzmann distributions. Despite the same functional form, this is in contrast to the standard case of statistical mechanics at equilibrium, where the link between molecular dynamics and the equilibrium distribution is non-

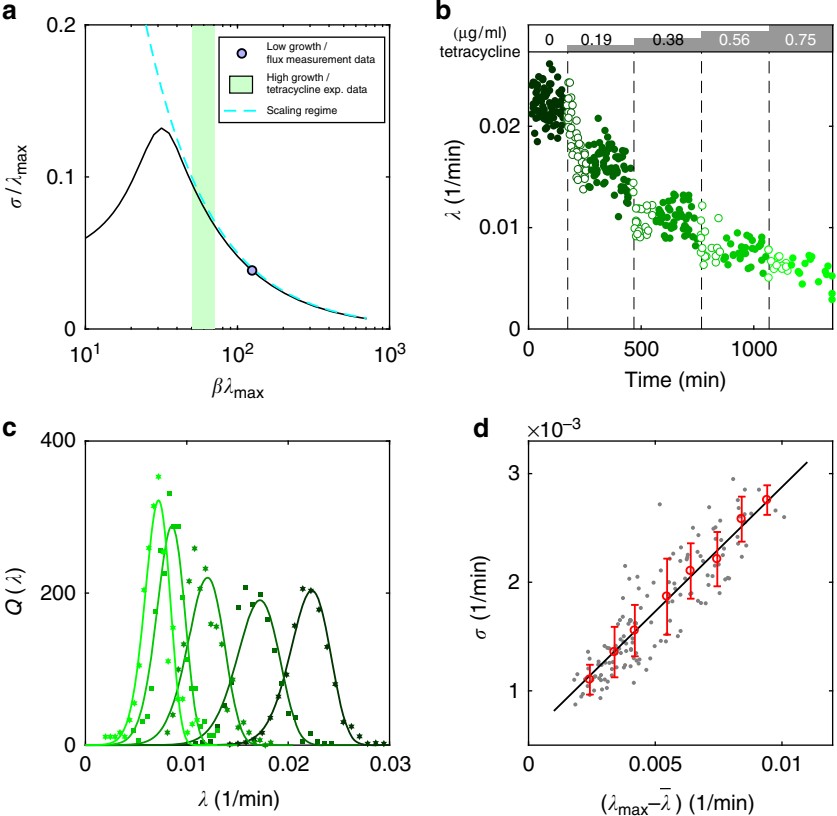

**Fig. 4 a** Predicted fluctuations in the growth flux, $\sigma/\lambda_{max}$, as a function of $\beta\lambda_{max}$. Model inferred from data in Fig. 2a is shown as a blue point and the approximate range experimentally probed in **b–d** is shown as green shade. In the scaling regime (cyan dashed line), $\sigma \propto (\beta\lambda_{max})^{-1}$. **b** single-cell measurements (each dot = one division event) of elongation rate in a microfluidic mother machine device for wild-type *E. coli* under increasing sub-inhibitory antibiotic concentrations, indicated on top; data from a single experiment in ref.[37]. After switching to higher antibiotic concentration (lighter shade of green), cells show transient behavior (empty green circles) which we ignore and focus only on steady state (full green circles). **c** Measured distributions (plot symbols) of growth rates from multiple experiments as in **b** (different shades of green = different tetracycline concentration). Solid lines show maximum entropy distributions with best-fit values for ($\beta$, $\lambda_{max}$) (Supplementary Note 3). **d** Testing the predicted scaling of growth rate fluctuations. Each dot = one lineage from experiments in **c**. The average growth rate, $\bar{\lambda}$, the maximal observed growth rate, $\lambda_{max}$, and the SD of the growth rates, $\sigma$, were estimated separately on each lineage. Scaling regime of **a** predicts a linear relationship, which is shown here by binned data (red points with errorbars = mean ± SD within equi-distant bins) and best linear fit (solid black line, $R^2 = 0.77$)

trivial. On the other hand, localization of growth distributions toward the optimum can also arise due to the active regulation of metabolic enzymes, most likely involved in catalyzing irreversible reactions. These two mechanisms are not mutually exclusive and can actually operate concurrently; in our estimate, the purely population-dynamics scenario substantially underpredicts growth rate optimization (i.e., $\beta\lambda_{\max}$) for the bulk culture, likely because it completely disregards active regulation of metabolism, which is known to be important. Note that the two mechanisms are, at least in principle, distinguishable experimentally: in the mother-machine device, there is no competition across the independent microfluidics channels, putting us in the regime with a very small $N_C/N_0$, which predicts smaller $\beta^*\lambda_{\max}$ than in the bulk, qualitatively in line with the observations. In other words, in bulk, both single-cell regulation and competitive growth may be active simultaneously, leading to higher growth rate optimization than in single-cell microfluidics measurements, where competitive growth is nearly absent. Further investigations are required to tease apart these two contributions quantitatively, in particular allowing for growth state transitions in modeling.

The connection between maximum entropy models and fluctuation-dissipation relations, Eq. (4), requires further assumptions that need to be tested separately, but makes a very strong prediction about the relationship between the autocorrelation time of growth fluctuations and the typical response time to, e.g., nutrient shifts. This relationship appears fundamental, since the response time is a central biological quantity measurable in bulk, while the fluctuation autocorrelations are microscopic, single-cell properties, which can be measured with recent experimental setups. Interestingly, the predicted response times lengthen with the degree of growth rate optimization, suggesting a trade-off between responsiveness to changes and efficiency in steady state; as a consequence, it is unclear whether the evolutionarily optimal outcome should be equated to complete growth rate optimization with no fluctuations, e.g., the FBA limit. Quantitatively, in stable environments where *E. coli* grows well and possibly achieves a high degree of growth rate optimization, one could experimentally look for signatures of long-timescale fluctuations, either directly in the growth signal, or by proxy through constitutive gene expression. Curiously, we report that the parameter $\beta$ of our model has the dimension of time, whose best-fit value inferred from *E. coli* data is of the order of 1 day.

Beyond extensions to dynamics, our analysis made two further theoretical contributions. First, it clarified the relative roles of stoichiometric constraints and the growth optimization assumption in FBA. The maximum entropy model is an explicit construction of a smooth interpolation between the uniform regime (where only stoichiometric constraints are active) and the FBA (where growth is maximized in addition). The uniform limit is a natural baseline—where no control is exerted by the cell—against which to compare the observed fluxes, their fluctuations, and correlations, as we have done in Fig. 3. Without this baseline comparison, it is hard to assess how surprising the observations of metabolic optimality should really be[3]. Our second theoretical contribution is the observation that a certain minimal information is needed to achieve a desired growth rate (Fig. 3g, h). This information is expressed in the same currency (bits) in which we measure the performance of regulatory networks, enabling us to suggest a trade-off that sets the optimal degree of metabolic control. Contrary to other cellular networks where estimation of information only has been done for single network components or simple pathways[36], the metabolic network is the sole case where we could estimate the lower bound on the required number of regulatory bits. Our statistical mechanics approach thus opens

a connection between metabolic networks and their regulatory counterparts, which is both of theoretical interest and could also be probed in comparative genomic studies.

## Methods

**General background**. We consider the set of reactions in the well-mixed, continuum limit. Let $S_{i\mu}$ be the stoichiometric coefficient of the metabolite $\mu$ (whose concentration is $c_\mu$) in reaction $i$, whose flux is $v_i$. The metabolic network dynamics is then given by mass balance equations:

$$\dot{c}_\mu = \sum_i S_{i\mu} v_i. \tag{5}$$

Assuming steady state, $\dot{c}_\mu = 0$, and including further constraints from thermodynamics, nutrient availability, and kinetic limits in the form of lower (LB) and upper (UB) bounds on fluxes, we obtain a convex polytope $\mathcal{P}$ of feasible steady states (metabolic phenotypes) in the space of fluxes:

$$\sum_i S_{i\mu} v_i = 0,$$
$$v_i \in \left[v_i^{\mathrm{LB}}, v_i^{\mathrm{UB}}\right]. \tag{6}$$

In addition to bona fide, well-balanced chemical reactions, constraint-based models often include a phenomenological biomass reaction in the form of a linear combination of metabolite fluxes, $\lambda(\mathbf{v}) = \sum_i \xi_i v_i$, where the proportions $\xi_i$ are set to mimic cell growth, i.e., the metabolite fluxes necessary to reconstitute the biomass of a new cell in a typical division time.

**The network**. The network employed in the study is the catabolic core of the genome-scale reconstruction iAF1260 (see Supplementary Methods), in a glucose-limited minimal medium in aerobic conditions[44]. The network comprises $N = 86$ reactions among $M = 68$ metabolites and includes glycolysis, pentose phosphate pathway, TCA cycle, oxidative phosphorylation, and nitrogen catabolism. The dimension of the resulting polytope $\mathcal{P}$ of allowed steady states is $D = 23$, from which we can efficiently draw flux configurations using Hit-and-Run Monte Carlo Markov Chain after suitable preprocessing[27] (see Supplementary Methods).

**Maximum entropy modeling**. FBA looks for the flux configuration $\mathbf{v}_{\max}$ that maximizes growth $\lambda_{\max} = \lambda(\mathbf{v}_{\max})$ subject to constraints given by Eq. (6), which can be easily found by linear programming. In contrast, our maximum entropy approach starts with a distribution over fluxes with a Boltzmann form, which assumes that the fluxes are as random as possible while achieving a desired average growth rate[26]:

$$P_\beta(\mathbf{v}) = \begin{cases} \frac{1}{Z} e^{\beta\lambda(\mathbf{v})} & \mathbf{v} \in \mathcal{P}, \\ 0 & \mathbf{v} \notin \mathcal{P}. \end{cases} \tag{7}$$

The parameter, $\beta$, of the distribution $P$ can then be set to match the predicted average growth rate to the measured growth rate, $\lambda_{\mathrm{data}}$:

$$\bar{\lambda}(\beta) = \int_{\mathbf{v} \in \mathcal{P}} d\mathbf{v}\, \lambda(\mathbf{v}) P_\beta(\mathbf{v}) = \lambda_{\mathrm{data}}. \tag{8}$$

Once $\beta$ is fixed, the joint distribution of Eq. (7) can be queried for average fluxes, flux correlations, or other quantities of interest that we discuss later.

The maximum entropy distribution with a constrained average growth rate has two interesting limits, as illustrated in Fig. 1. The growth rate, $\bar{\lambda}$, increases with $\beta$ (which we will refer to as an optimization parameter) until, in the limit $\beta \to \infty$, the distribution $P_\infty(\mathbf{v})$ collapses into a delta function at $\mathbf{v}_{\max}$, lying at the boundary of the polytope $\mathcal{P}$: this is the FBA solution that supports the maximal growth rate $\lambda_{\max}$. Conversely, as $\beta \to 0$, Eq. (7) yields a uniform sampling of fluxes over the permitted polytope $\mathcal{P}$: this uniform solution is an interesting baseline case for comparison because it incorporates all stoichiometric constraints but postulates no growth rate optimization. In statistical physics, high-$\beta$ regime (limiting toward the FBA solution) corresponds to the energy-dominated regime, while the low-$\beta$ regime (limiting toward the uniform sampling) corresponds to the entropy-dominated regime; the optimization parameter $\beta$ corresponds to the inverse temperature.

Apart from generic information-theoretic arguments put forward by Jaynes in support of the maximum entropy approach[20], are there further justifications for using the Boltzmann form in Eq. (7) that would be specific to the case of metabolic networks? Below, we consider two non-exclusive possibilities: active regulation at the single-cell level and competitive growth dynamics in a population.

**Information costs of regulation**. The first possibility is to mechanistically interpret the deviation of flux distributions away from the uniform sampling of the polytope $\mathcal{P}$ and its localization around the optimal solution as a consequence of the active regulation in the metabolic network. Such regulation could be achieved by, e.g., control over gene expression of key metabolic enzymes, or by allosteric feedback regulation mediated by metabolite concentrations or fluxes. The degree of localization of the flux distribution can be quantified by its entropy:

$$S(\beta) = -\int_{\mathbf{v} \in \mathcal{P}} d\mathbf{v}\, P_\beta(\mathbf{v}) \log P_\beta(\mathbf{v}). \qquad (9)$$

Because $P_\beta(\mathbf{v})$ is, by construction, a maximum entropy distribution with average growth rate $\bar{\lambda}(\beta)$, the decrease in entropy[26],

$$I(\beta) = S(\beta = 0) - S(\beta), \qquad (10)$$

is a measure for the minimal amount of information necessary to control the fluxes and achieve a given average growth rate. Equivalently, if we were to construct a regulation system that needs to realize the Boltzmann distribution of Eq. (7), $I(\beta)$ would provide a lower bound on its information demand. In the Results section, we estimate this information demand for the *E. coli* network and propose a toy regulatory model that can meet it.

**Competitive growth dynamics**. The second possibility is that the Boltzmann distribution emerges from competitive growth dynamics. Since its historical origins in statistical physics, much research has been devoted to uncovering the dynamical roots of Boltzmann distributions, whose study highlighted important concepts and applications, ranging from ergodicity to fluctuation-response relations. The same questions naturally arise in the context of its application in metabolism. It has been shown that the maximum entropy distribution at a fixed average growth rate is recovered independently and justified dynamically as the steady state of logistic growth[28]. Since the logistic growth is the standard model used to experimentally fit optical density curves[45], this link also provides a possible interpretation of the maximum entropy parameter $\beta$, as we discuss below.

Consider a population of initial size $N_0$ in a medium with carrying capacity $N_C$ and assume that the intrinsic growth rates of individuals, $\lambda_i$, are sampled independently from a distribution $q(\lambda)$, defined over the feasible polytope $\mathcal{P}$. In the simplest setting, upon neglecting growth state transitions, the number $n_i$ of cells with growth rate $\lambda_i$ will evolve in time according to

$$\frac{\dot{n}_i}{n_i} = \lambda_i \left(1 - \frac{N}{N_C}\right) \quad , \quad N(t) = \sum_{i=1}^{N_0} n_i(t). \qquad (11)$$

Then, $n_i(t) = e^{\beta(t)\lambda_i}$, with

$$\beta(t) = t - \frac{1}{N_C} \int_0^t N(t') dt'. \qquad (12)$$

Under a mean field approximation[28], the steady states of these dynamics are distributions with maximum entropy form at a fixed average growth rate, where the asymptotic optimization parameter, $\beta^\star$, is given implicitly by the equation

$$\int q(\lambda) e^{\beta^\star \lambda} d\lambda = \frac{N_C}{N_0}. \qquad (13)$$

Equation (13) can be viewed as a relationship between quantities that can be independently estimated for a specific experimental setup: the inoculum size ($N_0$) and carrying capacity ($N_C$) on the one hand, as well as the typical value of $\beta$, via Eq. (8) or direct fitting of measured metabolic fluxes, on the other.

Taken together, the two mechanisms, active regulation and competitive growth dynamics, need not be exclusive, and can operate concurrently. A simple diagnostic that could provide insight into the relative importance of both mechanisms is to examine whether the relationship of Eq. (13) is satisfied. If it were, it would suggest that the Boltzmann distribution is dynamical in origin. If, on the other hand, the values of $\beta$ inferred from fitting the maximum entropy model were higher than those derived from the $N_C/N_0$ ratio and Eq. (13), additional active regulation may be at work. In the Results section and Supplementary Methods, we provide estimates of these quantities for the experiments under consideration.

**Code availability**. We have provided in doi:10.15479/AT:ISTA:62 a C++ code implementing the Lovasz preprocessing as well as the Hit-and-Run algorithm and the polytope representation of the metabolic network used in this study. Please refer to the README.txt file for further information.

**Data availability**. The metabolic network employed in this study is the catabolic core from the genome-scale reconstruction iAF1260 and it is available in the Supplementary materials of the published reconstruction work[44]. The experimental estimates of the metabolic fluxes can be retrieved from the database[46] doi: 10.1093/nar/gku1137 (see also the Supplementary Methods and Supplementary Data 1). Single-cell growth rate data are available from the Supplementary materials published in[37].

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

## Acknowledgements

We acknowledge the support of the Austrian Science Fund grant FWF P28844 (G.T.) and of the People Programme (Marie Curie Actions) of the European Union's Seventh Framework Programme (FP7/2007-2013) under REA grant agreement no. [291734] (D. D.M).

## Author contributions

D.D.M. and G.T. conceived the study and developed the theory. D.D.M. performed the simulations. A.A. and D.D.M. carried out the data analysis. C.G. and T.B. carried out the experiments. All authors contributed in writing the final manuscript.

## Additional information

**Competing interests:** The authors declare no competing interests.

