## [Peer Review File · Nature Communications]

Reviewers' comments:

Reviewer #1 (Remarks to the Author):

Referee Report on *Statistical mechanics for metabolic networks during steady state growth*, by Daniele De Martino et al

I. SUMMARY OF THE PAPER

In the paper submitted, the authors generalize the method of FBA using maximum entropy models inspired by techniques from statistical mechanics, or more mathematically speaking, from classical information theory. The new methods and the results they show I find them extremely interesting, pushing forward the current knowledge and mathematical tools available in the area, and therefore worth publishing in Nature Communications. Besides, I have found the paper a real pleasure to read.

I have however some minor comments the authors need to answer before moving forward to publication.

Minor Comments

- Previous works that introduced mathematical techniques coming from statistical mechanics (in particular from spin glasses) to study metabolic networks seem to be missing. I understand this is not a review on the "use of statistical mechanics on metabolic networks", however I believe it should be important to cite those works to put the reader into the context on how previous works have tried to generalise FBA. I trust that the authors agree they should incorporate the following references:
 1. An analytic approximation of the feasible space of metabolic networks, A Braunstein, AP Muntoni, A Pagnani, Nature Communications 8 2017
 2. Fast inference of ill-posed problems within a convex space, J Fernandez-de-Cossio-Diaz, R Mulet, Journal of Statistical Mechanics: Theory and Experiment 2016 (7), 073207
 3. A Novel Methodology to Estimate Metabolic Flux Distributions in Constraint-Based Models, F Alessandro Massucci, F Font-Clos, A De Martino, I Pérez Castillo, Metabolites 3 (2013), 838-852
 4. A weighted belief-propagation algorithm to estimate volume-related properties of random polytopes, F Font-Clos, FA Massucci, IP Castillo, Journal of Statistical Mechanics: Theory and Experiment, P11003 (2012)
 5. Identifying essential genes in Escherichia coli from a metabolic optimization principle, C Martelli, A De Martino, E Marinari, M Marsili, IP Castillo, Proceedings of the National Academy of Sciences 106 (8), 2607-2611 (2008)
- On the same line, it is clear to me that the intended audience are biologist still using as a tool FBA. However, from a mathematical viewpoint, and considering the above references, I would be tempted to say that reference [1] above provides currently the state-of-the-art method for an efficient algorithm to perform uniform sampling. As such I would like to ask the authors the painful task of reconsidering Figure 2A on their paper to include as well the results using the uniform sampling method presented in [1]. I believe this will send a clear message on the mathematical power of the method presented here. Alternatively, this should be equivalent to using the results the authors already have from the Hit-and-Run algorithm, and get from there the "flux relative to GLC uptake" to incorporate in Figure 2A. Notice that I understand that this information is somewhat already summarised in Figure 2B, but I would prefer also to have the case $\beta = 0$ presented in figure 2A. Interestingly, the method introduced in [1] provides a powerful way to implement the method of maximum entropy presented here, without the need of scanning the polytope \mathcal{P} with Hit-and-Run, the latter being sometimes a numerically cumbersome task. I guess this is an idea worth pursuing for a future publication.

Reviewer #2 (Remarks to the Author):

In their manuscript "Statistical mechanics for metabolic networks during steady state growth", De Martino et al introduce an extension of FBA that accounts for variability in cell populations and assumes that the flux distributions in a bacterial population follow a Boltzmann distribution, with biomass production rates playing the role of a negative energy function. They validate several predictions from this model and present two interpretations of the Boltzmann distribution. In one interpretation, the "natural" distribution would be uniform, and thanks to regulatory mechanisms (which require cellular investments), the distribution becomes narrower and obtains its Boltzmann-type shape. In the other interpretation, the Boltzmann distribution itself arises naturally from the growth dynamics in the population.

The manuscript is very clearly written. The idea of a Boltzmann distribution on the flux polytope is, at least for me, a very intuitive thought, and I'm impressed by the experimental evidence that is given in the manuscript.

There is one major point I would like to stress. I was happy to see that the Boltzmann distribution is interpreted in two ways, as the result of an active regulation (with regulation cost), and as the result of growth dynamics (without any need for active regulation). However, if the Boltzmann distribution stems from growth dynamics, there is no need for regulation, no regulation cost, etc, and a whole part of the manuscript seems to be obsolete. I personally find the population dynamics explanation more convincing. Therefore, it should be made clear that these are two interpretations that may have their merits in their own rights, but that are actually mutually exclusive. (Please correct me if I'm mistaken with this point). Maybe a safe way to introduce the regulation idea would be to say: "If we were to construct a regulation system that needs to realise this Boltzmann distribution, its information demand would be ...")

Minor remarks (in random order):

"Today, one can infer nearly complete metabolic networks from genome-scale data,"
How can we know our inferred network are complete (unless we assume we know the "true" network)? For example, there may be a vast number of unspecific side reactions (with low fluxes) that we do not know. Maybe a less strong statement would be enough.

The word "fluctuations" seems to be used as a synonym for "variability". A fluctuation, at least in my use of the word, is something temporal. The manuscript is only about (timeless) steady states (except for the paragraph on population dynamics, where one could argue that population variability is in fact caused by temporal fluctuations). Therefore "variability" may be a better term in most places than "fluctuation"

It might be interesting to briefly discuss the "growth/yield" problem (the fact that "classical" FBA, with an active bound on the carbon source influx, actually maximizes yield, not growth; whereas other methods, such as FBA with molecular crowding, actually maximize - and predict - growth rates). To me, it looks like the Boltzmann approach is somewhere in between and may handle both cases.

Classical FBA suffers from the fact that internal "futile" flux cycles cannot be resolved because they have no effect on biomass production (and are therefore "neutral" with respect to the optimality problem). The Boltzmann approach, I think, inherits this problem; I could imagine cases in which an internal flux cycle, with an ill-determined flux, give a high probability weight to flux distributions that allow for this cycle to be active. Can you comment on this point?

"homeostasis (i.e., steady state, $c \cdot \mu = 0$)"

I'm not sure that homeostasis and steady state are equivalent. I think, "steady state" is the point

here, and maybe "homeostasis" need not be mentioned.

Just out of curiosity: why did you use the terms "bona fide chemical reactions" and "bona fide metabolic fluxes"?

Sometimes, the usage of the term "growth rate" was not fully clear; maybe you can distinguish between "single-cell growth rate" and "population growth rate", at least in cases where it may be unclear?

Again, also just out of curiosity: can you imagine cases where an "inverse" Boltzmann distribution, favouring low instead of growth rates, would be of interest?

"We inferred the single parameter of our model and found the value of β * by constraining"
-> "and" is unclear .. does it mean ", that is," ?

Please explain "annealed approximation" at least briefly; what are the assumptions / simplifications applied? How does the distribution $q(\lambda)$ come into play? Does the population model consider something like "mutations", or only an initial distribution and (quantitative) selection for growth?

"the dependence of β on the carrying capacity of the environment (N_C), the initial size of the colony (N_0), and the probability distribution from which the inoculum was sampled ($q(\lambda)$),"
Can you describe this dependence intuitively (to give a sense what the value of beta actually means)?

Some FBA methods use thermodynamic constraints (relationships between fluxes and metabolite concentrations or chemical potentials) to narrow down flux directions. Would there be non-trivial ways (ie, not just predefining all flux directions) to account for such thermodynamic constraints?

Typo: "isocytate"

Once more, for curiosity: would it be possible to "retrofit" one flux distribution that is most similar to the mean fluxes obtained from the Boltzmann distribution, and are there any heuristic principles to get such a flux distribution directly, without actually sampling the Boltzmann distribution?

"These are mainly nitrogen and phosphate transport reactions, and to a lesser extent, MDH and PGI reactions; the latter two reactions are classified as reversible, so the predicted increase may have

thermodynamic rather than regulatory reasons."

This is overstated; the model does not describe thermodynamics, nor kinetics; it refers only to flux distributions, and an ad hoc assumption about their probability distribution.

"implying high levels of ammonia in the cell."

Metabolite levels are not modelled; maybe "high ammonia fluxes"?

"Among tightly controlled fluxes"

There's no control in the model

Fig 3A, Legend: "and the corresponding range for $\beta \lambda_{\max}$ "

Unclear to me; doesn't the STD emerge from one specific choice of $\beta * \lambda_{\max}$; so why would this range matter?

Fig3H, Legend: why "entails metabolic cost"

Why "metabolic"?

"but postulates no regulation-mediated growth rate optimization"

Sounds misleading, because a non-uniform distribution could also come from simple kinetics (without additional regulation)

Reviewer #3 (Remarks to the Author):

The Authors address a very interesting problem, namely understanding what type of information can be derived from an FBA-like approach to the description of metabolic networks. In particular they adopt a maximum entropy approach to improve over standard FBA predictions, and ask several questions about flux distributions, regulatory information, and scaling.

As I said the topic is very interesting, but I am afraid I don't find the paper of high enough standards to be published in Nature Communications.

My main criticism is that some of the material presented here is very much an extension of findings already published by the Authors themselves elsewhere, see for instance the scaling concepts and the dynamic population model presented already in De Martino, Capuani, De Martino, Phys. Biol. 13 (2016) 036005. For this reason, while the manuscript certainly contains some novel elements, it is not of the overall level expected in Nature Communications. I would suggest that either the Authors just focus on one/two relevant and completely novel questions, or they more clearly explain in what sense what they present here is novel with respect to their own previously published work.

For this reason I am sorry that I cannot recommend the publication of the manuscript in its current form.

Answer to Referee 1

We thank the referee for his/her positive review. The referee provided timely references on new sampling methods that will be very useful in extending the maximum entropy approach, and making it practically applicable to experimental modeling.

- *Previous works that introduced mathematical techniques coming from statistical mechanics (in particular from spin glasses) to study metabolic networks seem to be missing. I understand this is not a review on the “use of statistical mechanics on metabolic networks”, however I believe it should be important to cite those works to put the reader into the context on how previous works have tried to generalise FBA. I trust that the authors agree they should incorporate the following references[...]*

We thank the referee for the interesting references, which we have added to the revised manuscript (page 8, column 2, lines 36–47). The issue of uniform sampling under constraints is the essence of the computational task to be solved to implement maximum entropy, and is known to be a difficult problem (e.g., in comparison to flux-balance analysis, FBA, which is based on simple linear programming). For the network that we focus on in our manuscript, the Markov Chain Monte Carlo method is efficient, but we now fully acknowledge in the revised manuscript that the advanced methods suggested by the referee would be preferable (faster) on larger networks and/or in performing inverse modeling (see also the reply to the second referee).

- *As such I would like to ask the authors the painful task of reconsidering Figure 2A on their paper to include as well the results using the uniform sampling method presented in (Braunstein et al). I believe this will send a clear message on the mathematical power of the method presented here. Alternatively, this should be equivalent to using the results the authors already have from the Hit-and-Run algorithm, and get from there the flux relative to GLC uptake to incorporate in Figure 2A. Notice that I understand that this information is somewhat already summarised in Figure 2B, but I would prefer also to have the case $\beta = 0$ presented in figure 2A.*

We have included in the new version of the manuscript the results of the uniform sampling in Figure 2A, alongside with FBA and maxent. The uniform sampling has been performed with the Monte Carlo for consistency with other results plotted in the same figure. On the network analyzed, the Monte Carlo does not suffer from any convergence problems.

Answer to Referee 2

We thank the referee for carefully reading the manuscript and for his/her constructive comments. The referee provided us with many insights that really improved the manuscript. The referee raises many interesting open-ended issues (touching upon ergodicity, thermodynamic constraints implementation, growth rate/yield trade offs, “negative temperature”, etc.) that will deserve further investigations and testify to the rich potentiality of a maximum entropy approach in modeling metabolic networks.

- *There is one major point I would like to stress. I was happy to see that the Boltzmann distribution is interpreted in two ways, as the result of an active regulation (with regulation cost), and as the result of growth dynamics (without any need for active regulation). However, if the Boltzmann distribution stems from growth dynamics, there is no need for regulation, no regulation cost, etc, and a whole part of the manuscript seems to be obsolete. I personally find the population dynamics explanation more convincing. Therefore, it should be made clear that these are two interpretations that may have their merits in their own rights, but that are actually mutually exclusive. (Please correct me if I'm mistaken with this point). Maybe a safe way to introduce the regulation idea would be to say: "If we were to construct a regulation system that needs to realise this Boltzmann distribution, its information demand would be ..."*

We thank the referee for raising this interesting point and we apologize if presenting two possible interpretations raised confusion. We do not believe that these explanations are mutually exclusive, but in comparing them we report that the growth dynamics argument, in its simple form presented in the manuscript, is not yet providing correct quantitative predictions, as we argue below.

First, the existence of mechanisms that actively regulate metabolism and growth is an established fact. Many such mechanisms have been demonstrated in *Escherichia coli*, from genetic mechanisms of attenuation to end-product allosteric inhibition in balancing amino acid production (a classical example being tryptophan, see Schleif, R., *Genetics and molecular biology*, Ed. 2., Johns Hopkins University Press, 1993), and the incorporation of such mechanisms into systems-level models is the focus of intense research. From this point of view, the minimum entropy reduction calculation in our model gives a rather conservative but reliable estimate of the amount of information a regulation system needs to provide in order to achieve the selected growth rate. As a further support of this view, we highlight in the new version of the manuscript that the number of out-of-equilibrium reactions (39), that are in general thought to be sites of possible metabolic regulation, matches well the number of required bits needed to achieve the observed experimental growth rate (40).

Second, we previously argued (*Phys. Rev. E* **96**: 010401, 2017) that a logistic growth model, extended in a simple way to populations with heterogeneous metabolic phe-

notypes, converges to a maximum entropy distribution at fixed average growth rate. Specifically, growth distributions from cells in mother machine are well described by the Boltzmann distributions with high values of β , i.e., cells are in principle able to reduce the entropy in the metabolic space without active regulation, as modeled by the logistic growth model. While the emergence of Boltzmann distribution from competitive growth is an interesting hypothesis, we believe that this explanation does not yet provide a quantitative description of the growth dynamics. The predictions of the heterogeneous logistic growth model for the lag phase (defined from the parameters of the logistic curve itself) and dependence on inoculum size/carrying capacity are only in qualitative agreement with experiments, with significant quantitative differences (reported in the revised manuscript, also see below) remaining between β inferred from logistic growth and directly from fitting flux predictions or the average growth rate.

It is entirely possible that both mechanisms, regulation and logistic growth, contribute to the emergence of Boltzmann distributions and that the discrepancy in the inferred β from the logistic growth model is precisely because regulation in that mechanism is ignored. We also note that, at least in principle, the two hypotheses can be experimentally distinguished: in the mother-machine device, cells are grown in separate microfluidic channels and thus do not really compete for the shared resource, but grow independently. The question is then whether, nevertheless, the Boltzmann distribution over fluxes is observed in single cells. If so, this suggests that regulation plays a role. While metabolic fluxes at the individual cell level are difficult to measure, the distribution of individual cell growth rates from the mother machine follows the maximum entropy predictions, as we report, *although these cells were not sampled from the same population grown in a batch culture, where competitive growth would have taken place* (i.e., in the setup most relevant for the logistic growth model).

Despite these arguments against the logistic growth mechanism, we believe that it represents an interesting theoretical observation: it can be compared to an analogous issue for standard equilibrium thermodynamic systems where the Boltzmann distribution emerges through more involved routes (ergodic hypothesis, Liouville theorem and Gibbs construction). Here, it could be ascribed to the exponential character of growth laws. However, in order to provide quantitative predictions, the population dynamics model should be suitably extended by including regulation. This aspect deserves further investigation, but is beyond the scope of our manuscript.

We have rewritten the manuscript to clearly point out the two mechanisms by which Boltzmann distributions can be established, emphasizing that they are not exclusive, and including the arguments outlined above.

- *Minor remarks (in random order):*

*"Today, one can infer nearly complete metabolic networks from genome-scale data,"
How can we know our inferred network are complete (unless we assume we know the*

"true" network)? For example, there may be a vast number of unspecific side reactions (with low fluxes) that we do not know. Maybe a less strong statement would be enough.

We agree with the referee on this point and have corrected our statement (page 1, column 1, lines 8–10).

- *The word "fluctuations" seems to be used as a synonym for "variability". A fluctuation, at least in my use of the word, is something temporal. The manuscript is only about (timeless) steady states (except for the paragraph on population dynamics, where one could argue that population variability is in fact caused by temporal fluctuations). Therefore "variability" may be a better term in most places than "fluctuation"*

We agree with the referee on this point with regard to metabolic fluxes, and have therefore changed the term into "variability" throughout the manuscript appropriately. In referring to growth rate variability we continue to use however interchangeably the term "fluctuations" and "variability" to keep the connection with physical literature, where these can be truly fluctuations in time as measured by monitoring the temporal traces of growth rate in mother machine experiments. We also point out an interesting underlying issue about ergodicity in bacterial populations, as would be naturally suggested by a statistical-mechanics-inspired approach, which is possibly connected with the findings of *Phys. Rev. Lett.* **108**: 238105 (2012).

- *It might be interesting to briefly discuss the "growth/yield" problem (the fact that "classical" FBA, with an active bound on the carbon source influx, actually maximizes yield, not growth; whereas other methods, such as FBA with molecular crowding, actually maximize - and predict - growth rates). To me, it looks like the Boltzmann approach is somewhere in between and may handle both cases.*

The flux data we have analyzed indeed refers to growth conditions below the acetate switch ($\bar{\lambda} < 0.4\text{h}^{-1}$) where classical FBA relies on yield maximization; more exactly, where the maximization of rate and yield are equivalent. To model metabolic overflows at higher growth rates, further constraints should be added. One of the strengths of the maximum entropy approach is that such extensions are very natural via a suitable modification of the energy function. For example, by including one more Lagrange multiplier (e.g., γ) to fix both average rate and yield, we would transform the exponent in the Boltzmann distribution as follows: $\beta\lambda \rightarrow \beta\lambda + \gamma\lambda/u$ (where u is the uptake). In the Discussion section of the revised manuscript we now mention this option (page 8, column 2, line 25), which, however, deserves a separate investigation.

- *Classical FBA suffers from the fact that internal "futile" flux cycles cannot be resolved because they have no effect on biomass production (and are therefore "neutral" with respect to the optimality problem). The Boltzmann approach, I think, inherits this problem; I could imagine cases in which an internal flux cycle, with an ill-determined flux, give a high probability weight to flux distributions that allow for this cycle to be active. Can you comment on this point?*

Internal closed thermodynamically unfeasible flux cycles can arise in models upon incorrect handling of thermodynamic constraints and, even if they have a neutral effect on biomass, they can lead to an incorrect scaling analysis (*Phys. Rev. E* **95**: 062419 (2017)) that is one of the points of our work. In the general case the maximum entropy approach does indeed inherit this problem whose full resolution leads to difficult computational issues (see below the reply to further remarks). In the model we analyzed (central carbon metabolism of *E. coli*) we have checked that no non-trivial thermodynamic inconsistencies are present using the methods defined in (*PLOS Comput Biol* **8**: e1002562, 2012; *Metabolites* **3**: 946–966, 2013). Only one trivial internal cycle of length two has been detected between fumarate reductase (FRD7) and succinate dehydrogenase (SUCDi) whose prescribed bounds are thus thermodynamically inconsistent. This has been corrected by considering a lumped reversible reaction with the same stoichiometry.

- *"homeostasis (i.e., steady state, $c\mu = 0$)" I'm not sure that homeostasis and steady state are equivalent. I think, "steady state" is the point here, and maybe "homeostasis" need not be mentioned.*

We agree with the referee on this point and have removed the use of homeostasis (page 2, column 1, line 26).

- *Just out of curiosity: why did you use the terms "bona fide chemical reactions" and "bona fide metabolic fluxes"?*

The term has been used to point out the phenomenological character of the biomass reaction in comparison with the other well-balanced enzymatic reactions.

- *Sometimes, the usage of the term "growth rate" was not fully clear; maybe you can distinguish between "single-cell growth rate" and "population growth rate", at least in cases where it may be unclear?*

We agree with the referee on this point. In the manuscript, we now explicitly use "population growth rate" wherever appropriate to avoid ambiguity when we refer to experimentally measured average population growth rate.

- *Again, also just out of curiosity: can you imagine cases where an "inverse" Boltzmann distribution, favouring low instead of growth rates, would be of interest?*

This can be done if one defines "negative temperatures," $\beta < 0$, and has been considered theoretically in *Phys. Rev. E* **96**: 010401 (2017).

- *"We inferred the single parameter of our model and found the value of β^* by constraining". Here, "and" is unclear .. deos it mean ", that is," ?*

Yes, that was the intended meaning. We amended the manuscript appropriately (page 4, column 1, line 7).

- *Please explain "annealed approximation" at least briefly; what are the assumptions / simplifications applied? How does the distribution $q(\lambda)$ come into play? Does the population model consider something like "mutations", or only an initial distribution and (quantitative) selection for growth?*

The annealed approximation amounts to substituting the average of a function for the function of the average, i.e. $\overline{f(x)} \simeq f(\overline{x})$. In statistical mechanics of disordered systems it is usually intended for the free energy, eg $\log \overline{Z} \simeq \overline{\log Z}$ (where Z is the partition function and the average is over disorder). Annealed approximation usually holds for so-called self-averaging extensive quantities for which fluctuations vanish in the thermodynamic limit. In such a case the system is said to be symmetric under replica exchange, see, e.g., *Spin glass theory and beyond: An Introduction to the Replica Method and Its Applications*, Vol. 9, World Scientific Publishing, 1987; Derrida, B., *Random-energy model: Limit of a family of disordered models Phys Rev Lett* **45**: 79, 1980. Full quenched solutions in disordered models are usually computationally difficult to obtain.

For the model under exam, we can verify that the law of large numbers holds for the number of cells N in the limit $N_0, K \rightarrow \infty$, $K/N_0 \rightarrow \text{const}$. This follows in fact the equation

$$N = \sum_{i=1}^{N_0} e^{\beta \lambda_i}, \quad (1)$$

where the λ_i are independent random variables identically distributed with the density $q(\lambda)$. Its average over disorder is thus (up to a factor) the moment generating function (Laplace transform) of $q(\lambda)$

$$\overline{N} = N_0 \int d\lambda q(\lambda) e^{\beta \lambda} \quad (2)$$

we have for the average of the square ($Z(\beta) = \int d\lambda q(\lambda) e^{\beta \lambda}$)

$$\overline{N^2} = N_0(N_0 - 1)Z^2(\beta) + N_0Z(2\beta) \quad (3)$$

Upon considering the approximation $q(\lambda) = (a + 1)(1 - \lambda)^a$ and $\beta > a$ we can compute the relative fluctuations

$$\frac{\overline{(N - \bar{N})^2}}{(\bar{N})^2} \simeq \frac{1}{N_0} \left(\frac{\beta^{a+1}}{(a + 1)!} - 1 \right) \quad (4)$$

that vanishes in the aforementioned limit $N_0, K \rightarrow \infty$, $K/N_0 \rightarrow \text{const.}$, since β is finite and given asymptotically ($t \rightarrow \infty$) by the equation $Z(\beta) = K/N_0$. However such convergence can be very slow given the large prefactor and finite size corrections shall be worked out in future studies, possibly employing recent techniques developed in the the field of statistical mechanics of disordered systems (see, e.g., Fyodorov et al. *Large time zero temperature dynamics of the spherical $p=2$ -spin glass model of finite size*. Journal of Statistical Mechanics: Theory and Experiment **2015.11** (2015): P11017) These calculations have been included in the revised version of the supporting information text.

- *”the dependence of β on the carrying capacity of the environment (N_C), the initial size of the colony (N_0), and the probability distribution from which the inoculum was sampled ($q(\lambda)$),” Can you describe this dependence intuitively (to give a sense what the value of beta actually means)?*

In essence, Eq (11) mathematically expresses that **(i)** the optimal high-growth, low-entropy regions in the metabolic phenotypic space can be reached by simple growth dynamics, but that **(ii)** this process is limited by the capacity of the environment and the size of the initial seed.

The integral in Eq (11) can be computed analytically in the approximation $q(\lambda) = (a + 1)(1 - \lambda)^a$ leading to

$$\frac{N_C}{N_0} = \frac{(a + 1)e^\beta \gamma(a + 1, \beta)}{\beta^{a+1}} \quad (5)$$

where $\gamma(a + 1, \beta)$ is the lower incomplete Gamma function. For $\beta > a \gg 1$, we have approximately

$$\log \frac{N_C}{N_0} \simeq \beta - (a + 1) \left(\log \frac{\beta}{a + 1} + 1 \right). \quad (6)$$

It can be seen that, for $\beta \gg a$ very large, if we consider an “equivalent” population growing at the maximum growth rate λ_{max} , β is the time (in units of λ_{max}^{-1}) where the carrying capacity and the steady state are reached $N_C \simeq N_0 e^{\beta \lambda_{max}}$ (upon restoring units). We now report the values of β calculated from this formula for a typical value in batch culture ($N_C/N_0 \simeq 10^6$) and extrapolated for the value in a mother machine ($N_C/N_0 \simeq 4$) as, respectively, $\beta \simeq 50$ and $\beta \simeq 20$ whereas we infer from experiments $\beta \simeq 100$ and $\beta \simeq 66$. Thus the simple logistic model underestimates β , but it correctly predicts that in batch cultures it is larger with respect to the mother machine, given the larger N_C/N_0 . We provide the above reasoning in the revised manuscript (page 6, column 2) and SI.

- *Some FBA methods use thermodynamic constraints (relationships between fluxes and metabolite concentrations or chemical potentials) to narrow down flux directions. Would there be non-trivial ways (ie, not just predefining all flux directions) to account for such thermodynamic constraints?*

The ways to implement thermodynamics without predefining flux directions are non-trivial since they require at least the existence of thermodynamic potentials and in turn avoid the presence of internal cycles (*Phys. Rev. E* **87**: 052108, 2013), leading to non-convex flux spaces (*Phys. Rev. E* **95**: 062419, 2017).

In such cases, Monte Carlo sampling (required to solve the maxent problem), and even linear optimization are, in general, unsolved computational issues. We briefly mention this issue in the revised manuscript. For the specific implementation in realistic metabolic network models, the amount of thermodynamic information gathered from experiments could possibly lead to difficult yet solvable examples. We are currently actively working on such examples.

- *Typo: "isocytrate"*

The typo has been corrected in the new version of the manuscript.

- *Once more, for curiosity: would it be possible to "retrofit" one flux distribution that is most similar to the mean fluxes obtained from the Boltzmann distribution, and are there any heuristic principles to get such a flux distribution directly, without actually sampling the Boltzmann distribution?*

We are not sure if we understand the question properly. But it is possible to devise a maximum entropy model where measured experimental fluxes are matched directly to the average of a maximum entropy distribution $\langle f_i \rangle = f_{i,exp}$. Such maximum entropy model would have the form $P(v) \propto e^{\sum_i h_i v_i}$, with a specific "field" h_i corresponding to every flux v_i whose average value the model should match to the measurements. It is computationally difficult but plausible to perform the inverse model and infer h_i for all constrained fluxes, especially when using advanced sampling techniques suggested by the first referee (referred in the new version of the manuscript at page 8, column 2, lines 36–47).

- *"These are mainly nitrogen and phosphate transport reactions, and to a lesser extent, MDH and PGI reactions; the latter two reactions are classified as reversible, so the predicted increase may have thermodynamic rather than regulatory reasons." This is overstated; the model does not describe thermodynamics, nor kinetics; it refers only to flux distributions, and an ad hoc assumption about their probability distribution.*

Our remark was intended to be read in the context of the hypothesis that out-of-equilibrium reactions in metabolism are putative sites of metabolic regulation. This is a current subject of research, but a well-known textbook example is glucose phosphorylation-dephosphorylation in glycolysis and gluconeogenesis. We have changed the sentence by reframing it and adding suitable citations (page 5, column 2, lines 3–6).

- *"implying high levels of ammonia in the cell." Metabolite levels are not modelled; maybe "high ammonia fluxes"?*

The referee is right that stoichiometric models do not model metabolite levels, but in general reaction fluxes are expected to be functions of concentrations and in such sense an inversion could give some insights on the concentration levels as well. In particular, for a reversible reaction, such as GLUDy, mass action law follows in the quasi-equilibrium approximation, and its inversion indicates a shift of products and substrates, either by increasing the former (indeed ammonia) or decreasing the latter (glutamate). Classic kinetic studies report quite a low affinity for ammonia $K_m \simeq 1\text{mM}$ (N. Sakamoto, A. M. Kotre, and M. A. Savageau, *Journal of bacteriology* **124**: 775–783, 1975), suggesting high level of ammonia .

- *"Among tightly controlled fluxes" There's no control in the model*

The term "control" has been removed in the new version of the manuscript.

- *Fig 3A, Legend: "and the corresponding range for \max " Unclear to me; doesnt the STD emerge from one specific choice of $\beta * \lambda_{\max}$; so why would this range matter?*

The experimental population value of $\bar{\lambda}/\lambda_{\max}$ comes from averaging over several experiments (with very similar dilution rates but different glucose uptakes and hence different λ_{\max} s) and thus it comes with an error (STD) that has to be statistically propagated when inferring the value of $\beta\lambda_{\max}$.

- *Fig3H, Legend: why "entails metabolic cost" Why "metabolic"?*

In general, regulation could come with energetic costs, e.g., energy expended by the cell to express (transcribe and translate) the required signaling proteins plus the energy expended for running the signaling networks (for instance, for running the phosphorylation and dephosphorylation cycles). In our simple toy model, the cost (expressed as growth rate reduction from zero-cost baseline) is assumed to be proportional to the number of signaling molecules required to control the metabolic network, which is why we referred to it as "metabolic" cost.

- *”but postulates no regulation-mediated growth rate optimization” Sounds misleading, because a non-uniform distribution could also come from simple kinetics (without additional regulation)*

The term “regulation-mediated” has been removed in the new version of the manuscript.

Answer to Referee 3

- *My main criticism is that some of the material presented here is very much an extension of findings already published by the Authors themselves elsewhere, see for instance the scaling concepts and the dynamic population model presented already in De Martino, Capuani, De Martino, Phys. Biol. 13 (2016) 036005. For this reason, while the manuscript certainly contains some novel elements, it is not of the overall level expected in Nature Communications. I would suggest that either the Authors just focus on one/two relevant and completely novel questions, or they more clearly explain in what sense what they present here is novel with respect to their own previously published work.*

We thank the referee for his/her review. We have taken under serious consideration the request to clarify the novel findings in our manuscript with respect to our previously published work, specifically *Phys Biol* **13**: 036005 (2016).

First, we clarify that our main contribution here is two-fold: we performed **(i)** the first comprehensive experimental test of the maxent framework for modeling metabolic networks, and **(ii)** suggested a number of extensions of this framework into new, experimentally relevant directions.

We provide for the first time compelling evidence that the maximum entropy framework actually works. The maximum entropy model has been confronted with high-dimensional experimental data (multiple fluxes, single-cell data) that it needed to fit simultaneously; this required substantial amounts of data analyses rather than just describing a qualitative match.

We carefully constructed a maximum entropy model for central carbon metabolism of *Escherichia coli* (for various technical nuances implicit in this task, see our responses to the second referee) and compared its flux predictions against experimental data. Maxent model significantly outperformed the standard flux balance analysis approach (Fig 2). We followed, also for the first time, in detail the behavior of specific enzymatic fluxes, averages and fluctuations and how pathways re-arrange as a function of growth optimization (Fig 3), highlighting a scaling behavior as a new prediction that we tested in subsequent experiments (Fig 4). We went beyond toy model constructions published in our previous work, and examined quantitative fits to data, reporting β values (and level of growth optimization) from batch cultures as well as single-cell microfluidic setup. We discussed two mechanisms for the emergence of the resulting Boltzmann distribution (through regulation and logistic growth dynamics) and argued that the second mechanism cannot fully account for quantitative data. Throughout the paper, we also suggest a number of theoretical extensions to the model that are worth pursuing in subsequent work, e.g., the simple regulatory pathway model to account for the needed regulatory information; suggestion to link fluctuations to mean response dynamics in future experiments;

modifications to maxent framework allowing yield maximization or inclusion of alternative constraints; etc.

Second, we report that the scaling referred to in the abstract of *Phys Biol* **13**: 036005, 2016, **is not** the scaling that has been proposed and verified in our new manuscript, but it is the usual scaling with respect to the mean highlighted in physical literature (e.g., *Phys Rev E* **93**: 012408, 2016). The focus of the mentioned paper is on the observation that single cell growth rate distributions could reflect heterogeneity of metabolic phenotypes. Results on the uniform sampling of metabolic models have been used to fit such distributions with the maximum entropy principle and by means of Lagrange multipliers. A detailed description of the metabolism was lacking as well as comparison with flux experiments. Intriguingly, reviewers for the *Phys Biol* paper acknowledged the proposal at the time as interesting and thought-provoking but *devoid of experimental evidence*. We view our current manuscript precisely as providing such evidence, for the first time.

Third, to clarify which contributions are novel in our submitted manuscript, we significantly reorganized the manuscript. Previously presented theory needed for the understanding of our paper has been summarized in a “Theoretical Background” section. Abstract and introduction have been modified to put emphasis on our aim to test the model against experimental data, and to suggest extensions to the maximum entropy model. We hope that the new structure clearly highlights new accomplishments of our paper and delineates them from our previous preparatory work.

Finally, we remark that in physics research it is very common that technical theory articles precede experimental work that verifies and extends them, the latter being considered much more relevant and thus worthy of publishing in high-impact journals, even at distance of many years (many famous examples can be given, from Higgs boson to the Landauer principle). Such a practice is starting to become fruitful in biology as well. We report that the same publication decisions were taken even for Flux Balance Analysis (FBA) that our paper extends: experimental evidence validating FBA was judged worthy of publishing in Nature journals (RU Ibarra, JS Edwards, BO Palsson, *Nature* **420**: 186, 2002; JS Edwards, RU Ibarra, BO Palsson, *Nature biotechnology* **19**: 125, 2001) while the theoretical framework has been proposed many years before (e.g., RA Majewski, MM Domach, *Biotechnology and bioengineering* **35**: 732, 1990).

REVIEWERS' COMMENTS:

Reviewer #1 (Remarks to the Author):

Second Referee Report on *Statistical mechanics for metabolic networks during steady state growth*, by Daniele De Martino et al

Introduction

I have read carefully the reply to both referees and the changes in the manuscript, and, therefore, I believe that the paper can proceed for publication. Below I simply go through the corrections and/or replies of the authors to both referees.

Regarding answers to Referee 1

I follow the authors' reply to referee 1, as submitted point by point:

1. I am pleased with the changes
2. I am pleased with the changes

Regarding answers to Referee 2

I follow the authors' reply to Referee 2, as submitted point by point:

1. I am pleased with the reply and the corresponding changes done in the manuscript.
2. I am pleased with the reply and the corresponding changes done in the manuscript.
3. I am pleased with the reply and the corresponding changes done in the manuscript regarding the use of "variability and fluctuations".
4. I am happy with the corrections.
5. I am pleased with the reply regarding internal "futile" flux cycles.
6. I am OK with the removing of the word "homeostatis"
7. Thanks for clarifying the use of the word "bona fide"
8. Thanks for clarifying between "population growth rate" and "single-cell growth rate"
9. I am please with the comment about effective negative temperatures.
10. I am happy with the amendment.
11. Regarding the annealed approximation: This is not entirely correct, although I understand this might be difficult to explain to non-(physicists from spin glasses and disordered systems). To talk about annealed approximation, one needs two sets of "dynamical variables". Annealed approximation then corresponds to treating the statistics of them on the same statistical footing. Your example $\overline{f(x)} \simeq f(\bar{x})$ is rather a mean-field approximation, which then you justify when computing the relative fluctuations given by Eq. (4) in your reply.
12. I am pleased with the reply and the corresponding changes done in the manuscript.
13. I am pleased with the reply.
14. Typo corrected.
15. I am pleased with the reply.
16. I am pleased with the reply and the corresponding changes done in the manuscript regarding *...nitrogen and phosphate transport reaction...*
17. I am pleased with the reply *high levels of ammonia*.

18. I am happy with the removal of the term *control*.
19. Please with the reply.
20. Pleased with the reply regarding *metabolic cost*.
21. Pleased with the removal of *regulation-mediated*.

Thank you for your correspondence regarding our manuscript “*Statistical mechanics for metabolic networks during steady state growth*”.

We thank the referees and we are pleased that they consider our manuscript ready for publication. The referees are satisfied with our replies and changes apart from one last remark that we address below.

- *Regarding the annealed approximation: This is not entirely correct, although I understand this might be difficult to explain to non-(physicists from spin glasses and disordered systems). To talk about annealed approximation, one needs two sets of dynamical variables. Annealed approximation then corresponds to treating the statistics of them on the same statistical footing. Your example $\overline{f(x)} \simeq f(\overline{x})$ is rather a mean-field approximation, which then you justify when computing the relative fluctuations given by Eq. (4) in your reply.*

We thank the referee for the remark. We agree with the referee and changed the term “annealed” with “mean field” in the new version of the manuscript. We have been using the term “annealed” in order to point out the analogy with the Random Energy Model, but in our case the random variables in question, respectively, the metabolic fluxes and the growth rate, are in the same set from a dynamical viewpoint upon assuming a steady state.